


# Time-dependent entrainment of smoke presents an observational challenge for assessing aerosol–cloud interactions over the southeast Atlantic Ocean

Michael S. Diamond[1], Amie Dobracki[2], Steffen Freitag[2], Jennifer D. Small Griswold[3], Ashley Heikkila[3], Steven G. Howell[2], Mary E. Kacarab[4], James R. Podolske[5], Pablo E. Saide[6,7], and Robert Wood[1]

[1]Department of Atmospheric Sciences, University of Washington, Seattle, Washington, USA.
[2]Department of Oceanography, University of Hawai'i at Mānoa, Honolulu, Hawai'i, USA.
[3]Department of Atmospheric Sciences, University of Hawai'i at Mānoa, Honolulu, Hawai'i, USA.
[4]School of Earth and Atmospheric Sciences, Georgia Institute of Technology, Atlanta, Georgia, USA.
[5]NASA Ames Research Center, Moffett Field, California, USA.
[6]Department of Atmospheric and Oceanic Sciences, University of California, Los Angeles, California, USA.
[7]National Center for Atmospheric Research, Boulder, Colorado, USA.

*Correspondence to*: Michael S. Diamond (diamond2@uw.edu)

**Abstract.** The colocation of clouds and smoke over the southeast Atlantic Ocean during the southern African biomass burning season has numerous radiative implications, including microphysical modulation of the clouds if smoke is entrained into the marine boundary layer. NASA's ObseRvtions of Aerosols above CLouds and their intEractionS (ORACLES) campaign is studying this system with aircraft in three field deployments between 2016 and 2018. Results from ORACLES-2016 show that the relationship between cloud droplet number concentration and smoke below cloud is consistent with previously reported values, whereas cloud droplet number concentration is only weakly associated with smoke immediately above cloud at the time of observation. Combining field observations, regional chemistry–climate modeling, and theoretical boundary layer aerosol budget equations, we show that the history of smoke entrainment (which has a characteristic mixing timescale on the order of days) helps explain variations in cloud properties for similar instantaneous above-cloud smoke environments. Precipitation processes are also expected to obscure the relationship between above-cloud smoke and cloud properties in parts of the southeast Atlantic, although marine boundary layer carbon monoxide concentrations for two case study flights suggest that smoke entrainment history drove the observed differences in cloud properties for those days. A Lagrangian framework following the clouds and accounting for the history of smoke entrainment and precipitation is likely necessary for quantitatively studying this system: an Eulerian framework (e.g., instantaneous correlation of A-train satellite observations) is unlikely to capture the true extent of smoke–cloud interaction in the southeast Atlantic.

## 1 Introduction

From June to October, fires spread across southern Africa produce more than a quarter of global carbon emissions from biomass burning (Roberts et al., 2009; van der Werf et al., 2010). The resulting smoke is frequently transported westward over



the southeast Atlantic Ocean (SEA) in association with the northern branch of the deep anti-cyclone over southern African and the Southern African Easterly Jet (Adebiyi & Zuidema, 2016; Garstang et al., 1996).

Low-level stratocumulus (Sc) clouds are abundant over the SEA due to strong lower tropospheric stability (LTS) from
subsidence and low sea-surface temperatures (Klein & Hartmann, 1993; Seager et al., 2003). The colocation of the plume of biomass burning aerosol (BBA) and clouds over the SEA has important radiative implications that depend on the vertical distribution of the smoke and clouds (Koch & Del Genio, 2010). The direct radiative effect of the smoke can be positive or negative depending on the underlying surface (Chand et al., 2009). If BBA is near Sc clouds, rapid cloud adjustments to the BBA direct effect, or semi-direct effects, can reduce cloud fraction (Ackerman et al., 2000), whereas smoke further aloft warms
the free troposphere (FT), increasing LTS and thus Sc cloud fraction and thickness (Sakaeda et al., 2011; Wilcox, 2010; Wilcox, 2012).

Recent observations of smoke aerosol in the boundary layer at Ascension Island during the Layered Atlantic Smoke Interactions with Clouds (LASIC) ARM Mobile Facility deployment make clear that smoke is mixing into the marine boundary
layer (MBL) in the SEA (Zuidema et al., 2018). When smoke mixes into the Sc clouds, a number of changes in cloud microphysical properties, or indirect effects, can result. For a given liquid water path (LWP), increasing the availability of aerosols that act as cloud condensation nuclei (CCN) increases the cloud droplet number concentration ($N_d$) and decreases the cloud effective radius ($r_e$): this "Twomey effect" increases the cloud albedo and thus produces a negative radiative forcing (Twomey, 1974). Rapid cloud adjustments to the Twomey effect can either enhance or counteract this negative radiative
forcing. For instance, the shift in the cloud droplet distribution toward smaller droplets may suppress precipitation (Albrecht, 1989); alternatively, the smaller droplets may evaporate more rapidly, increasing cloud-top entrainment and drying out the cloud (Wood 2007). This study will focus primarily on processes controlling the Twomey effect.

Previous observational work in this area has used the "A-train" constellation of satellites, which obtain data that is nearly
spatially and temporally coincident, to evaluate cloud response to BBA statistically. One method is to determine the slope of the logarithmic relationship between $N_d$, or $r_e$ if LWP is assumed fixed, and CCN (or a proxy like aerosol number concentration):

$$g = \frac{\partial \ln(N_d)}{\partial \ln(CCN)} = -3 \frac{\partial \ln(r_e)}{\partial \ln(CCN)}. \tag{1}$$

When clouds and smoke appear to be in contact, the linear slope of the logarithmic relationship between $r_e$ and the aerosol
index (AI), a proxy for aerosol concentration, has been estimated between -0.24 (Costantino & Bréon, 2010) and -0.15 (Costantino & Bréon, 2013), corresponding to values of 0.72 and 0.45 for the $N_d$–CCN relationship (g), within the range of previously calculated values for aerosol enhancement of $N_d$ (e.g., 0.71 in Kaufman et al., 1991; 0.5 in Nakajima et al., 2001). In contrast, $r_e$ and AI are uncorrelated when smoke and clouds are vertically well-separated.



Painemal et al. (2014) also find evidence suggestive of a measurable Twomey effect due to smoke in the SEA, although largely limited to the region north of 5° S. In this region, $r_e$ and cloud top height are anti-correlated. The authors interpret this as evidence that deeper clouds are more likely to be in contact with the overlying biomass burning layer, although the aerosol

base height as derived from the Cloud–Aerosol Lidar with Orthogonal Polarization (CALIOP) often shows separation between aerosol layer base and the cloud top height. Because the absorbing smoke particles attenuate the 532 nm CALIOP beam, the standard retrieval for aerosol base height is biased high, which may explain this discrepancy (Painemal el al., 2014; Rajapakshe et al., 2017).

Further complicating our understanding of the vertical distribution of BBA, models tend to show smoke subsiding rapidly over the SEA, whereas CALIOP observations show the plume staying at altitude for a much greater distance over the ocean (Das et al., 2017). The difficulty in reliably determining the lowest extent of the BBA plume is a large source of uncertainty regarding the strength and sign of BBA semi-direct and indirect effects over the SEA.

An implicit assumption made in the use of A-train observations is that in cases of smoke–cloud contact, the smoke is relatively well-mixed in the MBL at the time of observation. However, the process of cloud-top entrainment that mixes the FT smoke down into the MBL is not instantaneous. In idealized large eddy simulation models with smoke initially above clouds, it takes ~1–1.5 days after smoke–cloud contact for $N_d$ to level off at the CCN concentration of the smoke aloft (Yamaguchi et al., 2015; Zhou et al., 2017). Calculations of entrainment timescale presented below suggest these values may be toward the faster

end of what can be expected.

In this paper, we present new aircraft observations of clouds and BBA over the SEA region that show considerable variation in $N_d$ for very similar vertical distributions of BBA, calling into question the idea that MBL and FT BBA concentrations are in equilibrium. As a result, estimates of the magnitude of the radiative forcing from aerosol–cloud interactions ($RF_{ACI}$) due to

smoke over the SEA may be misleading without considering the transport history of the MBL air to assess for how long it has been entraining smoke. We suggest that failing to account for the relatively long timescale for entrainment – e.g., by using instantaneous correlations between above-cloud BBA and cloud properties – can obscure the true extent of microphysical modification of SEA stratocumulus by smoke.

## 2 Data and methods

### 2.1 ORACLES–2016 flights

The first deployment of the NASA ObseRvations of Aerosols above CLouds and their intEractionS (ORACLES) aircraft campaign, based out of Walvis Bay, Namibia (23.0° S, 14.5° E), took place during September 2016 (Zuidema et al., 2016).



ORACLES aims to characterize the aerosol–cloud system over the SEA throughout the biomass burning season; the second deployment, based out of São Tomé and Príncipe (0.3° N, 6.7° E), was completed in August 2017 and a third deployment is planned for October 2018. This study uses data acquired during the September 2016 field deployment (ORACLES-2016) from the P-3 Orion aircraft (P-3), a four-engine turboprop plane that can sample in situ from the top of the aerosol plume (~6 km

maximum) to ~100 m above the ocean surface.

All ORACLES-2016 science flights with valid data are included in this analysis: all P-3 in situ data used is available at espoarchive.nasa.gov. In addition, the 31 August (flight number PRF02-2016) and 4 September (PRF04-2016) flights are analyzed in greater detail as illustrative cases.

We use data from four specific flight maneuvers: ramps (RMP), in which the P-3 ascends or descends while continuing to travel horizontally; square spirals (SQS), in which the P-3 ascends or descends while spiraling over a fixed horizontal point; sawtooth legs (SAW), in which the P-3 porpoises through a cloud layer to sample air below, above, and within the clouds; and straight and level in–cloud legs (CLD).

For in-cloud legs, we accept data 5 minutes before/after the beginning/end of the leg for our above cloud (AC) and below cloud (BC) properties. We define AC properties as the mean value of a quantity between cloud top and 100 m above cloud top, adopting the 100 m value from Costantino & Bréon (2013) for satellite derived BBA–cloud contact. It should be noted that our AC values are only for the immediately above-cloud BBA and are not intended to be representative of aerosol higher

in the BBA plume. We define BC averages as the mean value of a quantity below 500 m. Although we expect most MBLs in our study area to be shallow and well-mixed, as is the case on both case study flights (31 August and 4 September), this introduces some uncertainty in the case of deeper, decoupled MBLs (Jones et al., 2011).

### 2.2 Cloud observations

Measurements of the cloud droplet number size distribution from 3 to 500 μm in diameter were made by an Artium Flight

Phase Doppler Interferometer (PDI) vertically mounted on a wing of the P-3 (Chuang et al., 2008). As droplets pass through the intersection of the PDI's two identical lasers, they act as lenses and refract light, producing a phase shift between the fringe patterns from the lasers that has a nearly linear dependence on droplet diameter. For further details on the PDI instrument and methodology, the reader is directed to Chuang et al. (2008).

We calculate $N_d$, $r_e$, and liquid water content (LWC) from the PDI's cloud droplet spectrum as follows:

$$N_d = \int_0^\infty n(r)dr \approx \sum_{i=1}^{128} n(r_i), \tag{2}$$





$$r_e = \frac{\int_0^\infty r^3 n(r) dr}{\int_0^\infty r^2 n(r) dr} \approx \frac{\sum_{i=1}^{128} r_i^3 n(r_i)}{\sum_{i=1}^{128} r_i^2 n(r_i)}, \tag{3}$$

$$LWC = \frac{4\pi}{3}\rho_w \int_0^\infty r^3 n(r) dr \approx \frac{4\pi}{3}\rho_w \sum_{i=1}^{128} r_i^3 n(r_i), \tag{4}$$

where $n(r)$ is the number of cloud droplets in a particular size bin, $r_i$ is the mean radius value for each of the PDI's 128 size bins, and $\rho_w$ is the density of liquid water. $N_d$ and $r_e$ averages are weighted by LWC; for $N_d$, this weighting reduces the impact of cloud edges to better represent the typical adiabatic cloud profile in which $N_d$ does not vary with altitude (Martin et al., 1994), whereas for $r_e$, this weighting emphasizes values higher in the cloud profiles, which are more comparable to that retrieved via satellite remote sensing (Nakajima & King, 1990). We then define a simple cloud mask, $N_d > 10$ cm$^{-3}$, that we apply before taking any averaged cloud or above and below cloud aerosol data. Mid-level clouds (defined here as any cloud observation above 3 km) are excluded from the analysis.

Remotely sensed $r_e$ and cloud optical thickness (COT) are retrieved by the NASA Langley Research Center from the Spinning Enhanced Visible and Infrared Imager (SEVIRI) aboard the geostationary Meteosat-10 satellite and $N_d$ is calculated assuming an adiabatic-like vertical stratification (Painemal et al., 2012; Painemal & Zuidema, 2011):

$$N_d = 1.4067 \times 10^{-6} \left[cm^{-\frac{1}{2}}\right] COT^{\frac{1}{2}} r_e^{-\frac{5}{2}}. \tag{5}$$

Only data from liquid clouds in the MBL (successful liquid cloud phase retrievals with effective temperatures below 280 K) are maintained for this analysis. For each flight analyzed, SEVIRI quantities are averaged over a 0.5° by 0.5° grid box centered at the P-3's location every 15 minutes. The flight average quantity is then the average of all the 15 minute values.

**2.3 Smoke and aerosol observations**

CCN concentrations at 0.3% supersaturation were measured by a Droplet Measurement Technologies CCN-100 continuous-flow streamwise thermal-gradient CCN chamber onboard the P-3 (Roberts & Nenes, 2005). Sulfate (SO$_4$) mass concentration was measured by an Aerodyne Aerosol Mass Spectrometer (AMS) operating in V-mode (Canagaratna et al., 2007). CCN and SO$_4$ measurements provide information about the total amount of hygroscopic aerosol available from sea spray, secondary production, and transport from the continent.

Refractory black carbon (rBC) from 53 to 524 nm mass equivalent diameter was measured using a Droplet Measurement Technologies single particle soot photometer (SP2) with a solid diffuser inlet outside the front cabin of the P-3 (Schwarz et al., 2006; Stephens et al., 2003). The SP2 uses laser–incandescence to identify refractory particles and was calibrated using fullerene soot effective density estimates from Gysel et al. (2011). More information about the laser-induced incandescence technique is provided by Stephens et al. (2003) and details of the SP2 in particular can be found in Schwartz et al. (2006).



Because rBC is formed by the incomplete combustion of organic material, it is an unambiguous indicator of non-marine aerosol (in our case, primarily smoke) and is accompanied by other combustion products, including carbon monoxide (CO) and organic aerosol (Bond et al., 2013; Shank et al., 2012).

In this study, we primarily use the rBC number concentration as a proxy for smoke concentration, bearing in mind the undercounting of rBC cores below 80 nm in diameter (Schwarz et al., 2010). We additionally use CO concentrations measured by an ABB/Los Gatos Research $CO/CO_2/H_2O$ Analyzer (Liu et al., 2017) as an indicator of smoke presence that is not affected by rapid removal processes like precipitation.

**2.4 Model output**

Trajectories initialized at 250 m (988 hPa) in the center of the P-3 flight track (15° S, 5° E) at 12 UTC for both the 31 August and 4 September flights were run backward isobarically for 5 days using the Hybrid Single Particle Lagrangian Integrated Trajectory Model (HYSPLIT) with Global Data Assimilation System meteorology on a 0.5° by 0.5° grid (Stein et al., 2015).

Data from forecasts of the Weather Research and Forecasting model (WRF) configured with aerosol-aware microphysics (AAM; Thompson & Eidhammer, 2014) used for flight planning during the ORACLES-2016 deployment are analyzed along the track of each trajectory to assess the transport history and degree of smoke interaction prior to sampling. WRF–AAM was configured similarly to Saide et al. (2016) with a 12 km resolution domain over most of Africa and the Atlantic using daily Quick Fire Emission Dataset (Darmenov & da Silva, 2015) biomass burning emissions constrained in near-real time with

satellite aerosol optical depth from the NASA neural network retrieval (Colarco et al., 2017). The forecasts include CO-tagged tracers for smoke emissions. The initial 24 hours of each daily forecast were combined to perform this analysis.

**3 Results**

**3.1 Relationship between above- and below-cloud aerosol and cloud microphysics**

Using data from all 13 ORACLES-2016 flights with measurements, cloud microphysical properties correlate well with CCN
and our smoke proxies in the MBL but poorly in the FT. Figure 1 shows mean $N_d$ plotted against mean above- and below-cloud CCN concentrations for all flight maneuvers with valid data. Means and 95% confidence intervals (parentheses) for the relevant parameters of all ordinary least squares (OLS) regressions (Seabold & Perktold, 2010) are determined via bootstrapping and are reported in Table 1. In the MBL, ln(CCN) and $\ln(N_d)$ correlate well, with a coefficient of determination ($R^2$) of 0.73 (0.50–0.89). The slope of the $\ln(N_d)$–ln(CCN) relationship, g, is 0.45 (0.31–0.60), in good agreement with the
previously estimated values discussed above. In contrast, the correlation in the FT seems surprisingly weak in light of the





previous A-train findings above that assume, to some degree, that aerosol–cloud contact means significant mixing, with an $R^2$ of 0.32 (0.01–0.74). The above-cloud $\ln(N_d)$–$\ln(CCN)$ slope, g, is 0.16 (0.02–0.30), considerably smaller than in the MBL and barely distinguishable from zero at the 95% confidence level.

To explore this apparent discrepancy further, we perform linear OLS regressions to predict $N_d$ using $SO_4$, which is a significant contributor to both marine and continental CCN, and rBC, which should serve as an unambiguous tracer of smoke. Figure 2 shows the results using (a) all variables (AC and BC $SO_4$ and rBC), (b) only AC and BC $SO_4$, (c) only AC and BC rBC, (d) only BC $SO_4$ and rBC, and (e) only AC $SO_4$ and rBC. $SO_4$ is a better predictor of $N_d$ than rBC alone, which is expected as $SO_4$ may be contributing to CCN from both "natural" marine and "polluted" continental sources whereas rBC is only a component
of a subset of the continental CCN, although the combination of the two adds predictive power. The decent correlation of rBC and $N_d$ provides evidence for the influence of smoky continental air on the marine cloud microphysical properties beyond changes in meteorology and marine aerosol sources.

     Interestingly, the regression using only the BC values of $SO_4$ and rBC is nearly as skillful as the full regression whereas the
regression using only the AC values has comparatively little skill. Moreover, although the coefficients for the regressions including both $SO_4$ and rBC are not reliable given their mutual correlation (and are provided primarily for the sake of reproducibility), the coefficients of the regressions using only $SO_4$ (Table 1, row 4) or rBC (row 5) also reveal that those regressions are driven by the BC values. This indicates that variability in aerosol properties immediately above the MBL has little immediate impact on the microphysics of the clouds below.

**3.2 Case study: Comparison of 31 August and 4 September flights**

     To illustrate the phenomenon of similar above-cloud aerosol profiles leading to different MBL properties, we focus on the two flights highlighted in Fig. 2: 31 August (reds) and 4 September (blues). Figure 3 shows the mean location and Table 2 reports the starting and ending latitude, longitude, and time for each flight maneuver analyzed. As can be seen both in Fig. 2 and the SEVIRI $N_d$ imagery in Fig. 3, the 31 August clouds had some of the highest $N_d$ observed in the ORACLES-2016 deployment,
whereas the 4 September clouds were on the lower-$N_d$ end of the spectrum.

     Figure 4 explores each flight maneuver on the two days in more depth, showing (a) vertical profiles (lines) of rBC and cloud top height (vertical placement of markers) for RMP and SQS legs and (b) the full cloud droplet spectra for CLD and SAW legs. Focusing first on the vertical smoke profiles, rBC concentrations were generally higher just above cloud top on 4
September than they were on 31 August, yet MBL concentrations of rBC were ~5 times greater on 31 August. Cloud properties (horizontal placement of markers) tell a similar story, with $N_d$ values from 31 August well above those from 4 September. Even within the profiles on 31 August, higher above-cloud rBC values do not necessarily correspond to higher $N_d$. Particularly





high $SO_4$ values on 31 August likely contributed to the incredibly high $N_d$ of some profiles (e.g., the ~700 cm$^{-3}$ observed for RMP1) but do not account for the difference in MBL rBC between the days.

There is an ~100 m "clear air slot" (Hobbs, 2003), or gap, between the bottom of the aerosol plume and cloud tops for RMP4 on 4 September, and a similar drop–off in smoke just above cloud for RMP2, but the RMP1 and RMP3 profiles for that flight show direct instantaneous contact. The narrow gap distance for RMP2 and RMP4 suggests the 100 m threshold for cloud–aerosol "contact" of Costantino & Bréon (2013) may exclude observations that the 250 m and 360 m thresholds of Costantino & Bréon (2010) and Rajapakshe et al. (2017), respectively, would inadvertently include as "mixed" cases.

For the CLD and SAW legs that allowed for more time in cloud, we show the averaged cloud droplet spectra (curves) along with average $N_d$ and $r_e$ (ticks) in Fig. 4(b). CLD1 and CLD2 immediately follow RMP1 and RMP3, respectively, on 4 September (Fig. 3), suggesting direct instantaneous smoke–cloud contact for those legs. Again, the 31 August flight shows much clearer evidence of MBL pollution, with droplet spectra shifted toward smaller drop sizes and higher concentrations, and thus higher $N_d$ and lower $r_e$, as compared with the 4 September values. This result is consistent with the SEVIRI $N_d$ values (stars), although SEVIRI $N_d$ is systematically lower than the in situ values for both days. The presence of overlying aerosol can create a low bias in remotely sensed COT without having a large effect on remotely sensed $r_e$ (Haywood et al., 2004; Wilcox et al., 2009), leading to an expected low bias in $N_d$.

Whereas the vertical profiles of BBA in Fig. 4(a) look fairly comparable, the WRF–AAM curtains along the HYSPLIT back trajectories shown in Fig. 5 reveal considerable variation in the histories of smoke–cloud contact between the two cases. The 250 m trajectories both originate in the Southern Ocean 5 days before sampling but differ markedly in the smoke environments they encountered before being sampled, as shown in the curtain plots of WRF–AAM biomass burning CO concentrations for 31 August in Fig. 5(b) and 4 September in Fig. 5(c). The MBL sampled on 31 August appears to have been in contact with smoke for several days beforehand, whereas the MBL on 4 September was overlain with clean air until ~1.5 days before sampling. Given the sharp gradient at the lower boundary of the smoke plume seen in both the observations and the model output, direct contact may have been even more limited. Observed CO (Fig. 6) is qualitatively consistent with the WRF–AAM output, with MBL average CO values on 31 August considerably above those from 4 September and among the highest seen during the deployment (all other flights shown in thin grey lines).



## 4 Discussion

### 4.1 Timescales for the entrainment of free tropospheric CCN

The importance of the different entrainment histories of the 31 August and 4 September cases, and implications for the SEA region more generally, can be illuminated using an idealized framework. Assuming no other source or sink terms besides FT

entrainment and that entrainment is in approximate balance with large-scale subsidence, the rate of increase in MBL CCN concentrations, $CCN_{MBL}$, for a constant exposure to a directly-above-cloud FT CCN concentration, $CCN_{FT}$, can be expressed as:

$$\frac{\partial CCN_{MBL}}{\partial t} = \frac{w_e}{z_i}(CCN_{FT} - CCN_{MBL}),\tag{6}$$

where $w_e$ is the entrainment rate, $z_i$ is the height of the MBL, and t is time (Wood et al., 2012). This equation has a characteristic

e-folding timescale ($\tau_{ent}$) for $CCN_{MBL}$ to equilibrate with $CCN_{FT}$:

$$\tau_{ent} = \frac{z_i}{w_e}.\tag{7}$$

For a typical entrainment rate of 0.4 cm s$^{-1}$ (Faloona et al., 2005; Wood & Bretherton, 2004) and MBL height of 1 km, the characteristic timescale is ~3 days for the CCN concentration in the MBL to reach equilibrium with FT levels. This estimate is in line with previous values of, e.g., ~4 days for the northeast Atlantic (Bretherton et al., 1995) and ~3 days for the tropical

Pacific (Simpson et al., 2014). Figure 7(a) shows that for a plausible range of $w_e$ from 0.2–0.7 cm s$^{-1}$ (Faloona et al., 2005) and $z_i$ from 500–1500 m, the characteristic e-folding timescale for entrainment mixing of MBL and FT air varies from approximately one day to one week.

To illustrate the effects of both differing entrainment mixing timescales and sampling at different times along the MBL

evolution, we conduct a thought experiment in which an MBL in equilibrium with a "clean" FT with $CCN_{FT} = 100$ cm$^{-3}$ is exposed to smoky FT air with $CCN_{FT} = 1000$ cm$^{-3}$ for three days, after which "clean" FT conditions return. Figure 7(b) shows the results of this scenario with an MBL with $z_i = 1$ km and a range of $w_e$ values. Three main features stand out: 1) for any given entrainment timescale, the strength of the aerosol–cloud interactions estimated from a single snapshot during smoke contact will depend heavily on the time of observation; 2) for any given point in time, the entrainment rate can cause up to a

factor of 2 difference in $CCN_{MBL}$; and 3) for all but the most rapidly entraining cases, MBLs remain more polluted 24 hours after exposure to smoke than they were after the first 24 hours of smoke exposure.

### 4.2 Effects of precipitation

The real situation in the SEA is more complicated than the equations presented here because it is unrealistic to expect $CCN_{FT}$ to remain constant over long time periods and large spatial gradients and precipitation/coalescence scavenging acts as a sink

for CCN that is unaccounted for above, among other issues. Precipitation, in particular, has been shown to be a primary driver





of regional and seasonal $N_d$ variability in subtropical Sc decks (Mohrmann et al., 2017; Wood et al. 2012) and even moderate amounts of drizzle can rapidly deplete an MBL of CCN (Wood, 2006).

To assess how the inclusion of precipitation processes affects the discussion of entrainment above, we adapt a fuller Lagrangian
MBL CCN budget equation from Wood et al. (2012) and Mohrmann et al. (2017):

$$\frac{D}{Dt}CCN_{MBL} = C\dot{C}N_{FT} + C\dot{C}N_{SS} + C\dot{C}N_{Growth} + C\dot{C}N_{Precip} + C\dot{C}N_{Dry}, \tag{8}$$

where the subscript *FT* refers to entrainment of air from the free troposphere (Eq. 6), *SS* to sea spray, *Growth* to growth in the
MBL from secondarily produced and other small particles to CCN-active sizes, *Precip* to precipitation/coalescence scavenging, and *Dry* to dry deposition. As in Wood et al. (2012) and Mohrmann et al. (2017), we eliminate the growth and dry deposition terms because of their uncertain formulations and negligible contributions to the total CCN budget.

Following Wood (2006), the loss of CCN due to coalescence scavenging is given by:

$$C\dot{C}N_{Precip} = -\frac{KP_{CB}h}{z_i}N_d, \tag{9}$$

where K (= 2.25 m² kg⁻¹) is a constant that depends on the collection efficiency of drizzle drops, $P_{CB}$ is the precipitation rate at cloud base, and h is the cloud thickness. This formulation assumes that the accretion of cloud droplets onto drizzle drops
(coalescence) is the primary sink of CCN, rather than non-activated MBL CCN being washed out by falling rain, which is true for the lightly drizzling Sc decks. Even if the drizzle does not reach the ocean surface, CCN are lost because thousands of cloud drops can be collected together and evaporate in the MBL to form one larger haze particle, conserving mass but depleting aerosol number. For an appropriate supersaturation, we can assume $N_d$ and $CCN_{MBL}$ are approximately equal.

To complete our CCN budget equation, we account for sea spray as:

$$C\dot{C}N_{SS} = \frac{F(\sigma)U_{10}^{3.41}}{z_i}, \tag{10}$$

where $F(\sigma)$ is a function of supersaturation and $U_{10}$ is wind speed at 10 m (Clarke et al., 2006; Wood et al. 2012). We assume a supersaturation of 0.3%, corresponding to $F(\sigma) = 214$ m⁻³ (m s⁻¹)⁻²·⁴¹, and a mean wind speed of 7 m s⁻¹, which is representative of the SEA.

We can now write the full Lagrangian CCN budget equation as:

$$\frac{DCCN_{MBL}}{Dt} = \frac{z_i}{w_e + KP_{CB}h}(CCN_{eq} - CCN_{MBL}), \tag{11a}$$





$$CCN_{eq} = \frac{CCN_{FT} + \frac{F(\sigma)U_{10}^{3.41}}{w_e}}{1 + \frac{KP_{CB}h}{w_e}},$$ (11b)

where $CCN_{eq}$, which accounts for the sea spray source and the precipitation sink, has taken the place of $CCN_{FT}$ from earlier. By adding precipitation, the equilibration timescale is reduced, as the timescales for FT entrainment and coalescence scavenging add in parallel:

$$\tau = \left(\frac{1}{\tau_{ent}} + \frac{1}{\tau_{Precip}}\right)^{-1} = \frac{z_i}{w_e + KP_{CB}h}.$$ (12)

Figure 8(a) shows the full equilibration timescale for the same range of $w_e$ as earlier and range of $P_{CB}$ from 0–1 mm day$^{-1}$, assuming $z_i = 1$ km and $h = 300$ m. Although the timescale is reduced with increasing drizzle, the magnitude remains on the order of days for the precipitation values experienced in the Sc decks.

Figure 8(b) shows the full CCN budget equation applied to a case with $CCN_{FT} = 1000$ cm$^{-3}$, $w_e = 0.4$ cm s$^{-1}$, $z_i = 1$ km, $h = 300$ m, and a range of $P_{CB}$ values. Unsurprisingly, as precipitation increases, the equilibrium level of $CCN_{MBL}$ is reduced regardless of how much smoke is present. However, the key features are qualitatively the same as in Fig. 7(b): 1) for any given precipitation rate, the strength of the estimated aerosol–cloud interactions will depend heavily on the time of observation; 2) for any given point in time, the precipitation rate can cause substantial differences in $CCN_{MBL}$; and 3) for light drizzle, MBLs remain more polluted 24 hours after exposure to smoke than they were after the first 24 hours of smoke exposure.

Heavily drizzle was not observed on the 4 September flight, but instantaneous daytime precipitation measurements would not be sufficient as an indication of coalescence scavenging in any case given that Sc drizzle tends to peak overnight (Smalley & L'Ecuyer, 2015). Additionally, the association of high (low) precipitation with low (high) $N_d$ suffers from ambiguous causality: the different precipitation rates may drive the $N_d$ values, but alternatively the $N_d$ values may drive the frequency and intensity of precipitation (i.e., precipitation suppression/lifetime effects). Without any additional information, it would be difficult to distinguish between the potential roles of precipitation versus entrainment history in explaining the vastly different MBL aerosol and cloud properties observed between 31 August and 4 September. Fortunately, for the ORACLES-2016 flights CO measurements can be invoked to resolve this ambiguity. Coalescence scavenging may have been the preferred explanation for the differences between 31 August and 4 September had the two days seen similar levels of MBL CO, which is not removed by precipitation processes. However, because MBL CO was much higher on 31 August than on 4 September (Fig. 6), the difference in smoke entrainment history is the most plausible cause of the differences in MBL aerosol loading and $N_d$.

**5 Summary and conclusions**

Data from the September 2016 deployment of the ORACLES campaign shows that the presence of smoke from biomass burning in southern Africa in the MBL is associated with cloud microphysical changes, but the presence of smoke near cloud





top has little association by itself with the cloud properties below. This finding is illustrated by two flights which have similar vertical distributions of BBA but markedly different MBL pollution levels. Model results suggest that the MBL air sampled on 31 August had been in contact with smoke for a considerably longer time period than that sampled on 4 September. We argue that considering the prior history of the smoke and MBL air is key to understanding the large variations between cases

with similar vertical profiles in the FT.

A serious treatment of the time-dependence of the entrainment process has a number of implications for studies that use a more instantaneous, or "Eulerian," viewpoint, such as the A-train studies reviewed above. For instance, because the climatological MBL flow is southerly in the SEA, an instantaneous snapshot of smoke–cloud contact in the southern reaches of the domain

may underestimate the microphysical effects by not accounting for their upstream manifestation. Similarly, apparently "clean" cases in the northern part of the domain may have been polluted downstream, complicating efforts to compare "mixed" and "unmixed" statistics.

Recent modeling work suggests that accurately characterizing $RF_{ACI}$ is important for both regional and global estimates of

radiative forcing: Lu et al. (2018) find that smoke over the SEA can produce a net -7–8 W m$^{-2}$ forcing, primarily due to the Twomey effect, which corresponds approximately to an appreciable -0.089 W m$^{-2}$ forcing globally during the biomass burning season. Previous LES modeling of the Sc to cumulus transition also suggested that aerosol–cloud interactions over the SEA could contribute to net negative radiative forcings (Yamaguchi et al., 2015; Zhao et al., 2017). Inaccurate observational estimates of the magnitude of aerosol–cloud interactions over the SEA thus can greatly hinder our understanding of the

magnitude and sign of the net radiative forcing of smoke over the SEA and how changes in southern African biomass burning may affect regional and global climate. Although fire activity in southern Africa has been increasing over the past decade in opposition to global trends of reduced burned area associated with anthropogenic land-use change (Andela et al., 2017), it is reasonable to expect that biomass burning may decrease in the future in response to concerns about the negative population health consequences of particulate matter due to fires (Johnston et al., 2012) and the possibility that smoke has been suppressing

precipitation on the continent (Hodnebrog et al., 2016). Therefore, an accurate estimate of the climatic effects from a changing BBA loading over the SEA is highly societally relevant.

Future work is needed to assess to what extent a Lagrangian framework (Eastman & Wood, 2016; Mauger & Norris, 2010) accounting for the transport history of both the smoke and clouds differs from the traditional Eulerian framework in terms of

estimated aerosol–cloud interactions. Of course, other sources and sinks of aerosols besides FT entrainment – e.g., precipitation – act on similar timescales and may be better understood in a Lagrangian framework as well. Combining observations with the history of air masses from models is likely necessary to understand MBL aerosol loading, and thus $RF_{ACI}$ and resulting cloud adjustments.



**Data availability**

All ORACLES-2016 in situ data used in this study are publicly available at espoarchive.nasa.gov.

**Acknowledgements**

ORACLES is funded by NASA Earth Venture Suborbital-2 grant NNX-15AF98G. We would like to thank the crew of the
NASA P-3 Orion and our ORACLES-2016 pilots Michael Singer, Mark Russell, and Scott Farley in particular for their
dedicated support and patience with our somewhat unorthodox flight requests. In addition, we thank NASA's Earth Science
Project Office for their invaluable support with field logistics and Jens Redemann for his steady leadership throughout the
campaign.

The authors gratefully acknowledge the NOAA Air Resources Laboratory (ARL) for the provision of the HYSPLIT transport
and dispersion model. We thank the SatCORPS team at NASA-LaRC (https://satcorps.larc.nasa.gov/) for the provision of
Meteosat-10/SEVIRI retrievals. We would like to acknowledge high-performance computing support from Yellowstone
(ark:/85065/d7wd3xhc) provided by NCAR's Computational and Information Systems Laboratory, sponsored by the National
Science Foundation. The National Center for Atmospheric Research is sponsored by the National Science Foundation.

Michael Diamond's work was supported by NASA Headquarters under the NASA Earth and Space Science Fellowship
Program, grant NNX-80NSSC17K0404, and by an American Meteorological Society Graduate Fellowship sponsored by
NASA Earth Science.

Gregory Carmichael's contributions in generating the WRF-AAM output are greatly appreciated. Nikolai Smirnow was
instrumental in collecting the rBC data used here. We thank Yohei Shinozuka for his stewardship of the ORACLES data and
the provision of merged instrument data files. We would also like to thank Adeyemi Adebiyi, Sarah Doherty, Siddhant Gupta,
Johannes Mohrmann, Samuel Pennypacker, and Arthur Sedlacek for their helpful comments and suggestions during the
preparation of this work and Paquita Zuidema for her detailed notes from the 4 September flight.

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



**Tables**

**Table 1. Coefficient of determination ($R^2$), regression coefficients ($\beta$), and intercepts for all OLS regressions. Values reported as means with 95% confidence interval in parentheses as determined via bootstrapping.**

| Row | Figure | y | $x_1$ | $x_2$ | $x_3$ | $x_4$ | $R^2$ | $\beta_1$ | $\beta_2$ | $\beta_3$ | $\beta_4$ | intercept |
|---|---|---|---|---|---|---|---|---|---|---|---|---|
| 1 | Fig. 1(a) | $\ln(N_d)$ | ln(CCN) | – | – | – | 0.73 (0.50–0.89) | 0.45 (0.31–0.60) | – | – | – | 3.08 (2.26–3.86) |
| 2 | Fig. 1(b) | $\ln(N_d)$ | ln(CCN) | – | – | – | 0.32 (0.01–0.74) | 0.16 (0.02–0.30) | – | – | – | 4.54 (3.67–5.39) |
| 3 | Fig. 2(a) | $N_d$ | BC $SO_4$ | AC $SO_4$ | BC rBC | AC rBC | 0.70 (0.51–0.85) | 76 (47–109) | -47 (-111–-2) | 0.82 (-0.18–2.33) | 0.24 (0.00–0.48) | 115 (77–156) |
| 4 | Fig. 2(b) | $N_d$ | BC $SO_4$ | AC $SO_4$ | – | – | 0.52 (0.26–0.74) | 74 (55–99) | 5 (-14–25) | – | – | 143 (95–196) |
| 5 | Fig. 2(c) | $N_d$ | BC rBC | AC rBC | – | – | 0.38 (0.12–0.63) | 2.10 (0.78–3.76) | -0.08 (-0.41–0.19) | – | – | 184 (141–226) |
| 6 | Fig. 2(d) | $N_d$ | BC $SO_4$ | BC rBC | – | – | 0.61 (0.40–0.79) | 63 (34–94) | 0.86 (0.05–1.80) | – | – | 121 (81–162) |
| 7 | Fig. 2(e) | $N_d$ | AC $SO_4$ | AC rBC | – | – | 0.16 (0.03–0.36) | 35 (-16–99) | 0.03 (-0.23–0.34) | – | – | 216 (171–260) |



**Table 2. Starting and ending latitude, longitude, and time for all the flight legs from the 31 August and 4 September cases used in Fig. 3, 4, and 6.**

| Name | Start latitude (°) | End latitude (°) | Start longitude (°) | End longitude (°) | Start UTC Time | End UTC Time |
|---|---|---|---|---|---|---|
| 31 August 2016 (PRF02-2016) | | | | | | |
| RMP1 | -22.3 | -21.5 | 12.5 | 11.5 | 08:27 | 08:46 |
| CLD1 | -17.8 | -18.2 | 7.6 | 8.0 | 13:58 | 14:07 |
| SQS1 | -17.2 | -17.2 | 7.0 | 7.0 | 13:14 | 13:45 |
| RMP2 | -17.5 | -16.5 | 7.3 | 6.2 | 10:06 | 10:26 |
| CLD2 | -15.7 | -15.2 | 5.5 | 5.0 | 10:40 | 10:50 |
| SAW1 | -13.9 | -14.9 | 3.7 | 4.7 | 12:14 | 12:35 |
| CLD3 | -13.4 | -13.6 | 3.2 | 3.5 | 12:04 | 12:09 |
| SQS2 | -12.8 | -13.0 | 2.6 | 2.8 | 11:33 | 11:55 |
| 4 September 2016 (PRF04-2016) | | | | | | |
| RMP1 | -19.9 | -19.2 | 9.9 | 9.1 | 08:50 | 09:03 |
| CLD1 | -18.8 | -18.3 | 8.6 | 8.1 | 09:12 | 09:21 |
| RMP2 | -17.7 | -16.9 | 7.5 | 6.7 | 09:33 | 09:49 |
| RMP3 | -14.7 | -14.2 | 4.5 | 4.0 | 10:29 | 10:38 |
| CLD2 | -13.7 | -13.2 | 3.5 | 3.1 | 10:48 | 10:56 |
| RMP4 | -12.6 | -11.9 | 2.4 | 1.8 | 11:09 | 11:22 |



**Figures**

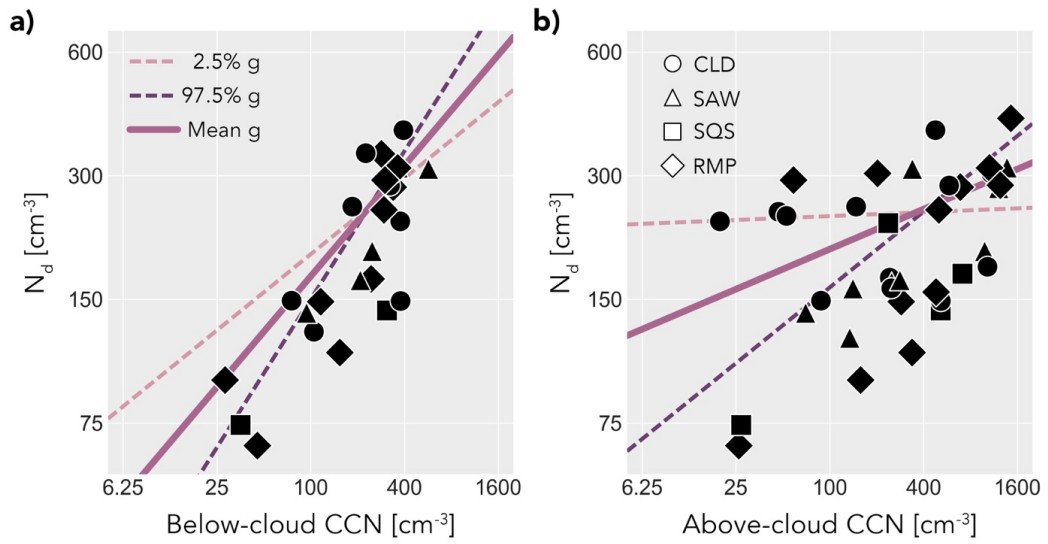

**Figure 1. Scatterplots of $N_d$ against (a) below-cloud and (b) above-cloud CCN concentration from all ORACLES-2016 flights. Solid and dashed purple lines show the mean value and 95% confidence interval of g, respectively.**



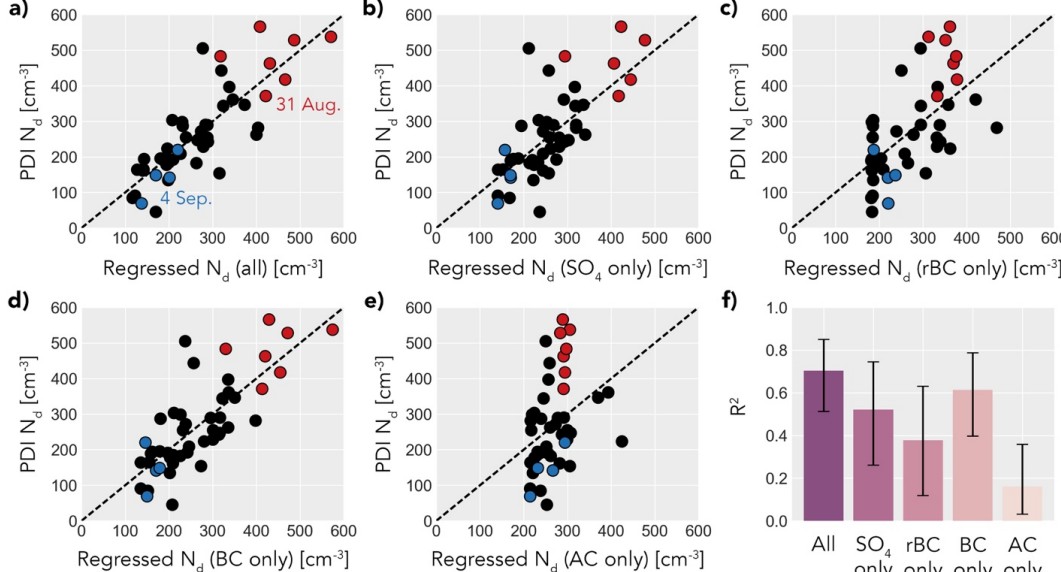

**Figure 2. Scatterplot of observed $N_d$ against $N_d$ predicted from a regression using (a) all valid AC and BC SO$_4$ and rBC observations, (b) only SO$_4$ observations, (c) only rBC observations, (d) only BC observations, and (e) only AC observations. 31 August and 4 September flights are highlighted in red and blue, respectively. Dashed black lines show the one-to-one line. (f) Bar chart showing $R^2$ for all regressions.**



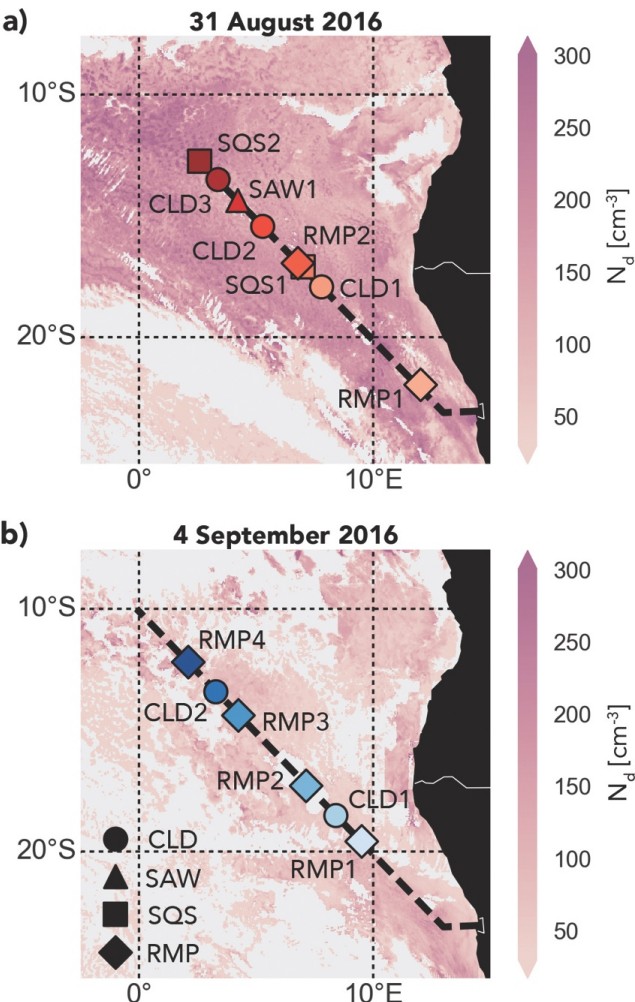

Figure 3. Map of the SEA region with the location of relevant flight legs for the (a) 31 August and (b) 4 September flights shown in
shades of red and blue, respectively. The flight track of the P-3 for each day is given by a dashed black line. Background shading is
$N_d$ from SEVIRI (12:15 UTC) screened for MBL clouds.





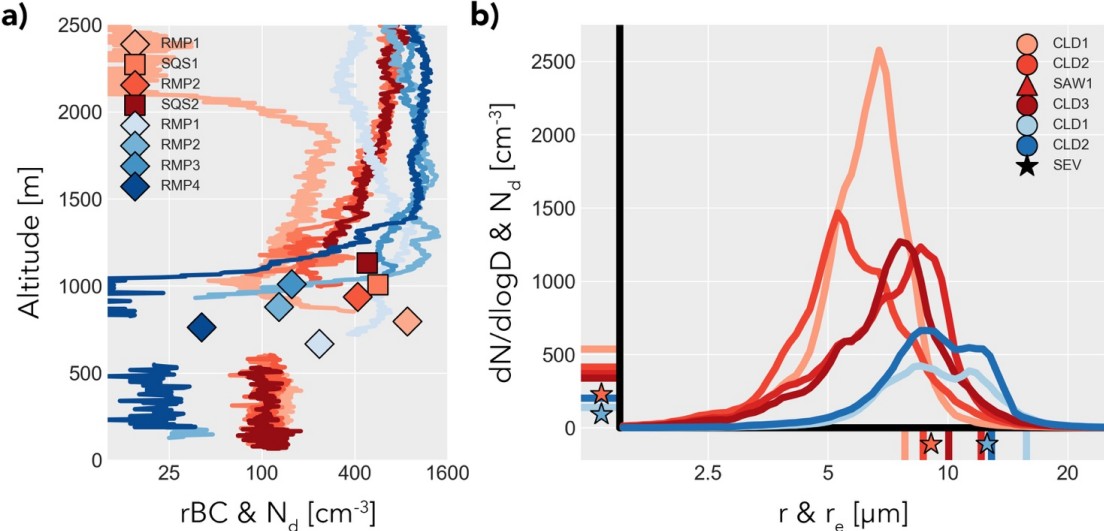

**Figure 4. Cloud microphysical properties and rBC for the 31 August (reds) and 4 September (blues) flights. (a) Vertical profiles of**
5   **rBC number concentration (lines) and average $N_d$ (markers) for each profile. Vertical marker position indicates cloud top height. Note that rBC and $N_d$ share the same x-axis because they have the same units and similar magnitudes. (b) Average cloud droplet spectra (curves), $N_d$ (ticks on y-axis), and $r_e$ (ticks on x-axis) for each CLD and SAW leg. Stars indicate the values of $N_d$ and $r_e$ from SEVIRI averaged over the 31 August (light red) and 4 September (light blue) flight paths.**





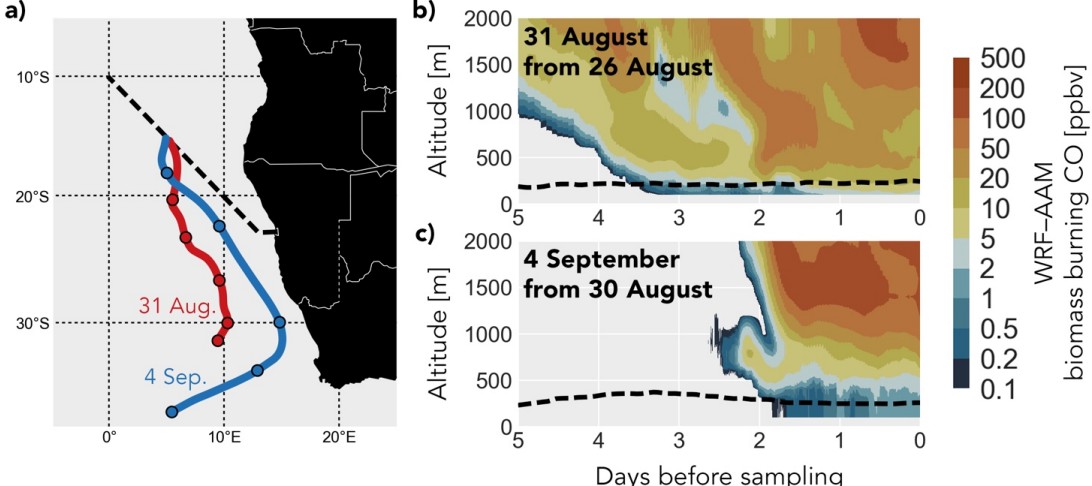

**Figure 5. (a) Map of HYSPLIT MBL back trajectories for 31 August (red) and 4 September (blue). Circles are plotted every 24
hours after initialization. Curtains of WRF–AAM biomass burning tagged CO along the path of the trajectories are plotted in (b)**
5 **for 31 August and c) for 4 September, with the trajectory altitude indicated by the dashed black line.**





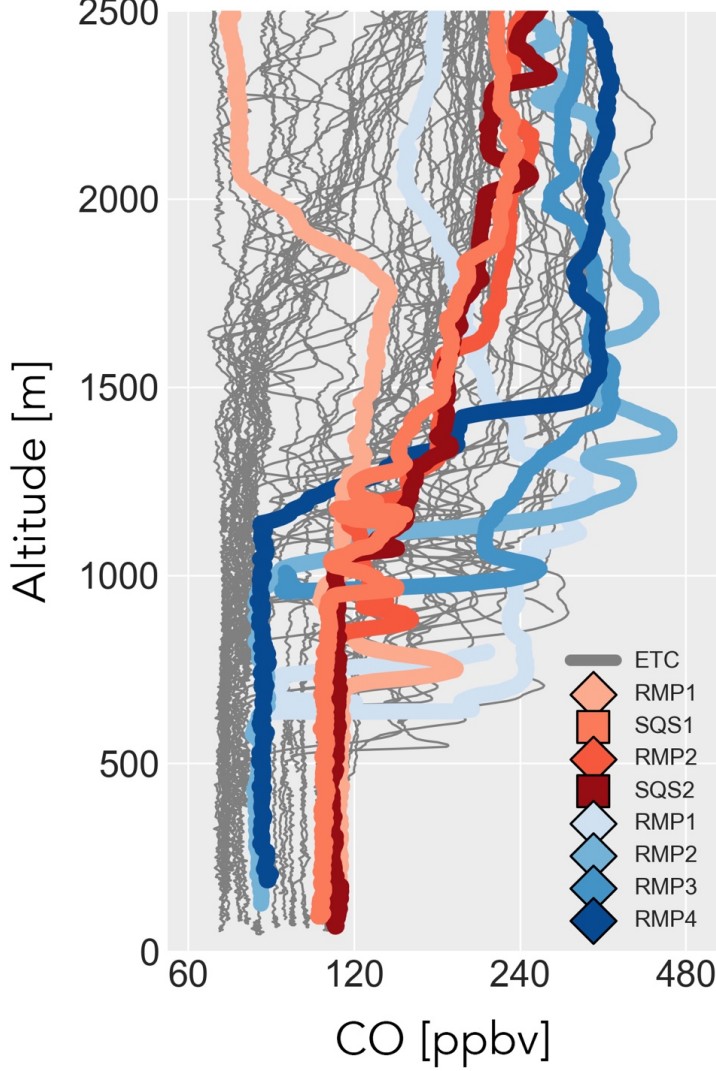

**Figure 6. Vertical profiles of observed CO for all ORACLES-2016 flights (grey lines), with the profiles from 31 August and 4 September highlighted in shades of red and blue, respectively.**





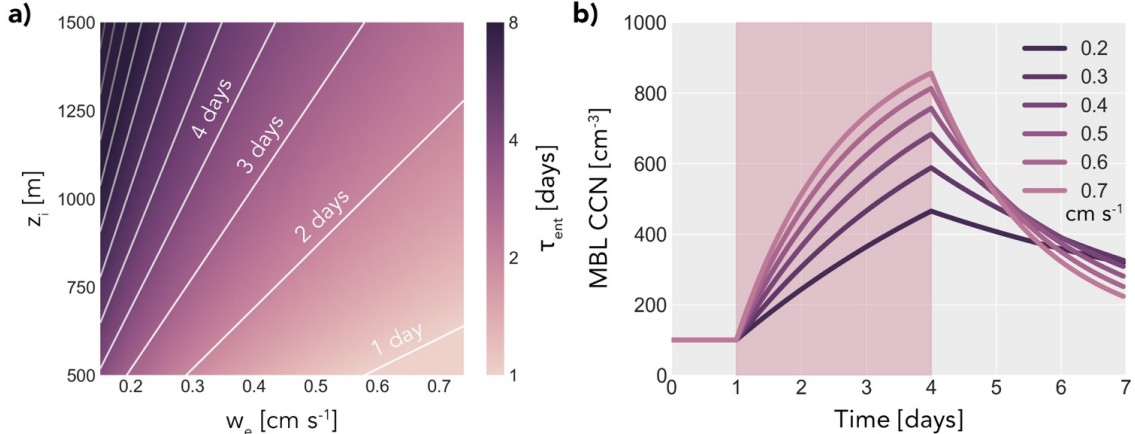

**Figure 7. (a) Characteristic (e-folding) entrainment mixing timescale for a range of plausible $z_i$ and $w_e$ values. Contours at one day**
5    **intervals for reference. (b) Evolution of $CCN_{MBL}$ over time in response to the introduction of a smoke plume with $CCN_{FT} = 1000$ cm$^{-3}$ at day 1 and its removal at day 4 (highlighted). Curves show results for a range of entrainment timescales with $z_i = 1$ km and $w_e$ varying between 0.2 and 0.7 cm s$^{-1}$.**





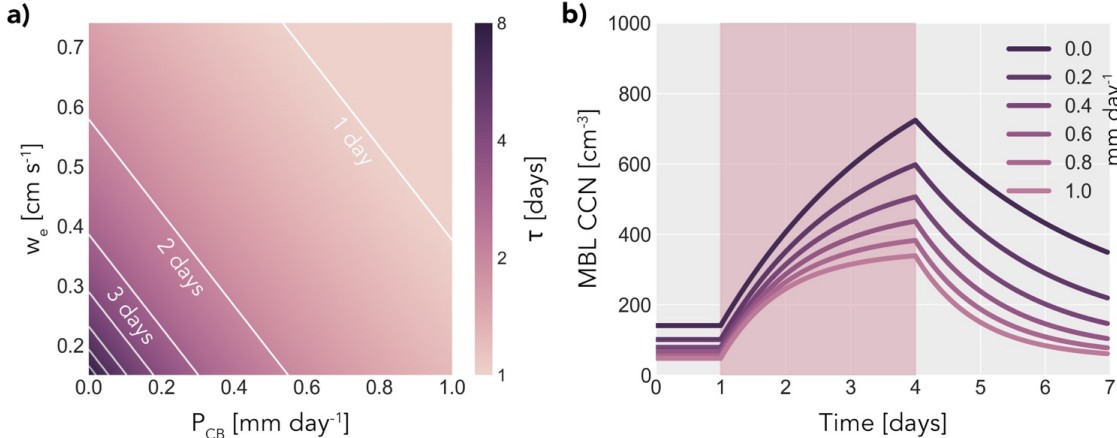

**Figure 8. (a) Characteristic (e-folding) entrainment mixing timescale for a range of plausible $w_e$ and $P_{CB}$ values. Contours at one–day intervals for reference. (b) Evolution of $CCN_{MBL}$ over time in response to the introduction of a smoke plume with $CCN_{FT} = 1000$ cm$^{-3}$ at day 1 and its removal at day 4 (highlighted). Curves show results for a range of equilibration timescales with $z_i = 1$ km, $w_e = 0.4$ cm s$^{-1}$, h = 300 m, and $P_{CB}$ varying between 0 and 1 mm day$^{-1}$.**