# Peer review of "Time-dependent entrainment of smoke presents an observational challenge for assessing aerosol-cloud interactions over the southeast Atlantic Ocean"

_Atmospheric Chemistry and Physics, 2018_

## Referee Comment (RC1) · Anonymous Referee #3 · 20 Jul 2018

General comments: This manuscript examines the relationship between cloud droplet number concentration and smoke below and above the cloud layer, using data from the September 2016 deployment of the ORACLES campaign over the southeast Atlantic Ocean. The results show that the smoke from biomass burning in the boundary layer is more strongly associated with cloud microphysical changes than that near cloud top. Using theoretical boundary layer aerosol budget equations, the authors show that the timescale for CCN in the boundary layer to equilibrate with CCN in the free troposphere is on the order of days, and the strength of the aerosol-cloud interactions

depends heavily on the time of observation, regardless of the drizzle rate. Together with back trajectory analysis from WRF-AAM and observation of marine boundary layer carbon monoxide concentrations, the authors conclude that smoke entrainment history is the key driver to the observed differences in cloud properties. The results are well presented and structured. The study is valuable for encouraging continued thought and discussion on accessing aerosol-cloud interactions over the southeast Atlantic Ocean. Recommendation: Acceptance with minor revisions

Main comment: The authors show a significant contrast between Nd vs. BC CCN and Nd vs. AC CCN relationships in Figs. 1 and 2. I am a little concerned if the better correlation between Nd and BC CCN is partly due to the different definitions of AC and BC properties. It is defined in the text(Page 4) that AC properties are 100 m averages while the BC properties are 500 m averages. I think it is worth mentioning when presenting results that the amount of data for averaging is different for BC and AC properties. Have the authors tried comparing 100 m averages for both AC and BC properties? How much difference is it in terms of $R2$? (The results probably would not be qualitatively different given well-mixed condition in the boundary layer.)

Specific comments: Page 2, Line 8-10: Potential edits: ....reduce cloud fraction (Hansen et al., 1997; Ackerman...) by reducing stability and relative humidity of the PBL, whereas ... (Johnson et al., 2004; Sakaeda ...)

Page 2, Line 22: add "for clouds with little precipitation" after cloud

Page 7, Line 5-12: Some of the conclusions seem to be drawn from Table 1. Add citation somewhere in this paragraph, or simply add the values of $R2$ to Figure 2.

Page 8, Line 1: add "(Fig. 2b)" after August

Page 8, Line 11-12: I am not sure I understand why the sequence of flight maneuvers suggests direct instantaneous smoke cloud contact

Page 12, Line 18: Zhao et al., 2017 is not in the reference.

---

## Author Comment (AC1) · 26 Jul 2018

Thank you for your review and constructive suggestions for clarifying the manuscript. We are incorporating your advice into the manuscript and will provide a detailed point-by-point reply in the Author's Response after we have had a chance to see all reviewer comments and short comments.

Before then, however, we would like to share a response to your main comment regarding the definition of the "below cloud" average and concerns about the below cloud

[Figure]

average taking into account more vertical data for averaging.

For our below cloud average, we use all non-cloudy data below 500 m: this sometimes yields data from 500 m to near-surface, but more often the P3 did not sample all the way to the near-surface. It would be difficult to restrict the below cloud averages to a 100 m vertical range consistently because the P3 did not always sample at the same altitude in the MBL. One idea could be to restrict the analysis to 100 m below cloud base, although the ambiguity in establishing the cloud base (which was generally not as well defined as cloud top and complicated by the occasional presence of shallow cumulus below the Sc decks) was what led us to establish the "below 500 m" definition originally. When the analysis for Figure 1a is performed using the "100 m below cloud base" definition (defining "cloud base" simply as the lowest altitude observation with Nd > 10 cm-3) the R2 is 0.62 and the slope is 0.43, not significantly different from the R2 of 0.73 and slope of 0.45 found in the original analysis. As pointed out above, and as can be seen in Figures 4 and 6, the MBL was generally well-mixed during ORACLES-2016 sampling, which does help explain why the results are fairly insensitive to the exact definition of the below cloud average. This may be a larger concern for follow-up analyses using ORACLES-2017, CLARIFY, and ORACLES-2018 data, as those campaigns observed decoupled MBLs much more frequently than ORACLES-2016 did.

We should also mention that despite generally averaging over a greater vertical distance, the exact number of 1 Hz observations that go into the averages is not necessarily greater for the below cloud than the above cloud leg because the P-3 may have spent more time just above cloud than below 500 m during a particular maneuver.

On the other end, we did experiment with increasing the vertical extent of the above cloud averages to 250 m and 360 m to better match the analyses of Costantino & Bréon (2010) and Rajapakshe et al. (2017), respectively. Correlations with Nd uniformly drop as the above cloud averaging distance increases for CCN, SO4, and rBC. As mentioned on page 8, lines 15-17, one explanation for this is that the coarser average

includes cases with narrow separations between the overlying aerosol and cloud tops.

---

## Referee Comment (RC2) · Anonymous Referee #2 · 3 Aug 2018

**Summary of manuscript**

The manuscript presents an analysis of South-East Atlantic in-situ data of boundary layer and free tropospheric biomass burning aerosol and of boundary layer cloud properties from the NASA ObseRvtions of Aerosols above CLouds and their intEractionS (ORACLES) campaign. The manuscript identifies the importance of distinguishing, in particular in satellite studies, whether or not biomass burning aerosol is merely present in the free troposphere (FT) or has entrained into the boundary layer (BL). This insight

is obtained by correlation analysis of boundary layer cloud drop number with aerosol concentration in the BL and the FT from all ORACLES flights, and by contrasting two ORACLES flights with high FT aerosol concentrations but different BL aerosol concentration and cloud drop number. The analysis is supported with satellite data, WRF-AAM simulations, and a calculation of the time scale for entrainment of FT aerosol into the boundary layer based on theoretical considerations. A consistent picture arises: Biomass burning aerosol in the free troposphere, in particular as seen by satellite, is a poor predictor of boundary layer cloud properties (specifically cloud drop number), because entrainment of aerosol from the free troposphere into the boundary layer is a comparably slow process (time scale of days). The mere presence of FT biomass burning aerosol determined in satellite observations above the boundary layer cannot be taken as an indicator of an effect on BL clouds, and accordingly, satellite studies that use "snapshots" of the state of the system may suffer from a bias in determining the effect of the biomass burning aerosol on boundary layer clouds. However, once biomass burning aerosol has entrained from the FT into the BL, it gives rise to a response of BL cloud properties that is consistent with established understanding. In conclusion, the entrainment history of a given BL air mass needs to be taken into consideration when studying biomass burning aerosol-cloud interactions, best by pursuing a Lagrangian framework that follows the air mass and accounts for entrainment of free tropospheric aerosol into the boundary layer.

The introduction gives a comprehensive but brief summary of interactions between biomass burning aerosol and boundary layer clouds in the South-East Atlantic and of pertinent work on the subject. The scope and key findings of the manuscript are summarized in the last paragraph. The data and methods section describes the ORACLES campaign and the instrumentation used, and gives a brief overview of the satellite data analysis and of the modeling approach. The results section presents analysis of aerosol and cloud data from all ORACLES flights, and contrasts two flights that highlight the key insights of the work. The discussion section elaborates on the theoretical approach to calculating the time scale for entrainment of aerosol from the FT into the

BL. The final section summarizes the elements of the previous sections, provides essential context, and lays out the conclusions.

**Summary of review**

The manuscript is compelling, clear, and concise, and presents relevant and new insights. It is in very good shape; some minor points require attention.

**Specific comments**

Page 1, line 23: "Precipitation processes are also expected to obscure the relationship between above-cloud smoke and cloud properties in parts of the southeast Atlantic, although marine boundary layer carbon monoxide concentrations for two case study flights suggest that smoke entrainment history drove the observed differences in cloud properties for those days."

This sentence could be simplified for clarity; e.g.

"Precipitation processes can obscure the relationship between above-cloud smoke and cloud properties in parts of the southeast Atlantic, but marine boundary layer carbon monoxide concentrations for two case study flights suggest that smoke entrainment history drove the observed differences in cloud properties for those days."

Page 4, line 7: "... in situ data used is available at ..."

replace with

"... in situ data used are available at ..."

Page 5, line 15: "Only data from liquid clouds in the MBL (successful liquid cloud phase

retrievals with effective temperatures below 280 K) ..."

Please specify what effective temperature is and adjust wording so that it is clear what "below" means (temperature or altitude).

Page 7, line 18: "This indicates that variability in aerosol properties immediately above the MBL has little immediate impact on the microphysics of the clouds below."

This seems to follow from Table 1, Row 7, rather than from the context of this sentence.

Page 8, line 26: "Observed CO (Fig. 6) is qualitatively consistent with the WRF-AAM output, with MBL average CO values on 31 August considerably above those from 4 September and among the highest seen during the deployment (all other flights shown in thin grey lines)."

It seems a bit too much to ask the reader to compare Fig. 5 b, c (WRF-AAM data) with Fig. 6 (ORACLES data) and to come to this conclusion. Please add, e.g., a time averaged vertical profiles of CO from Fig. 5 b, c to Fig. 6 to enable the comparison.

Page 12, line 7: "For instance, because the climatological MBL flow is southerly in the SEA, an instantaneous snapshot of smoke-cloud contact in the southern reaches of the domain may underestimate the microphysical effects by not accounting for their upstream manifestation. Similarly, apparently 'clean' cases in the northern part of the domain may have been polluted downstream, complicating efforts to compare 'mixed' and 'unmixed' statistics."

This sentence is a bit confusing. It seems to make more sense if "upstream" and "downstream" are switched.

---

## Short Comment (SC1) · 9 Aug 2018

Dr. Steven Abel of the UK Met Office has kindly pointed out that there is a typo in Eq. 11a. The fraction on the right hand side of the equation should be reversed: as written, this fraction is the characteristic e-folding timescale (Eq. 12), whereas it should be the inverse (as in Eq. 6). Figure 8, which uses Eq. 11a, was not affected by the drafting error.

-Michael Diamond

---

## Author Comment (AC2) · 27 Aug 2018

**Authors' Response to Reviewers**

**General comments:**

We would like to thank both reviewers for their thoughtful suggestions. We have revised the manuscript on the basis of this feedback and believe the revised product to be much clearer as a result. Specific changes and responses to the reviewer comments are outlined point-by-point below, with reviewer comments in red italicized font and the authors' response in black.

In addition to these changes, we have made an update to the data availability section (and a similar update in the Methods section and references) to reflect the fact that while the paper was undergoing review, the final DOI identifier became available for the ORACLES-2016 in situ data files (still provided by the NASA ESPO data archive). We have also expanded on the methods detailing how specific flight maneuvers were flagged (see revised page 4, lines 15-21) in response to several private inquiries and in further response to the main comment from Review 3 (our initial response reproduced below). Finally, Dr. Steve Abel has kindly pointed out in correspondence a typo in one of our equations — it has also been fixed in the revised manuscript (revised page 11, line 9).

**RC1 (Reviewer 3):**

*Main comment: The authors show a significant contrast between Nd vs. BC CCN and Nd vs. AC CCN relationships in Figs. 1 and 2. I am a little concerned if the better correlation between Nd and BC CCN is partly due to the different definitions of AC and BC properties. It is defined in the text(Page 4) that AC properties are 100 m averages while the BC properties are 500 m averages. I think it is worth mentioning when presenting results that the amount of data for averaging is different for BC and AC properties. Have the authors tried comparing 100 m averages for both AC and BC properties? How much difference is it in terms of R2? (The results probably would not be qualitatively different given well-mixed condition in the boundary layer.)*

For our below cloud average, we use all non-cloudy data below 500 m: this sometimes yields data from 500 m to near-surface, but more often the P3 did not sample all the way to the near-surface. It would be difficult to restrict the below cloud averages to a 100 m vertical range consistently because the P3 did not always sample at the same altitude in the MBL. One idea could be to restrict the analysis to 100 m below cloud base, although the ambiguity in establishing the cloud base (which was generally not

as well defined as cloud top and complicated by the occasional presence of shallow cumulus below the Sc decks) was what led us to establish the "below 500 m" definition originally. When the analysis for Figure 1a is performed using the "100 m below cloud base" definition (defining "cloud base" simply as the lowest altitude observation with $N_d > 10$ cm$^{-3}$) the $R^2$ is 0.62 and the slope is 0.43, not significantly different from the $R^2$ of 0.73 and slope of 0.45 found in the original analysis. As pointed out above, and as can be seen in Figures 4 and 6, the MBL was generally well-mixed during ORACLES-2016 sampling, which does help explain why the results are fairly insensitive to the exact definition of the below cloud average. This may be a larger concern for follow-up analyses using ORACLES-2017, CLARIFY, and ORACLES-2018 data, as those campaigns observed decoupled MBLs much more frequently than ORACLES-2016 did.

We should also mention that despite generally averaging over a greater vertical distance, the exact number of 1 Hz observations that go into the averages is not necessarily greater for the below cloud than the above cloud leg because the P-3 may have spent more time just above cloud than below 500 m during a particular maneuver.

On the other end, we did experiment with increasing the vertical extent of the above cloud averages to 250 m and 360 m to better match the analyses of Costantino & Bréon (2010) and Rajapakshe et al. (2017), respectively. Correlations with $N_d$ uniformly drop as the above cloud averaging distance increases for CCN, SO$_4$, and rBC. As mentioned on page 8, lines 15-17, one explanation for this is that the coarser average includes cases with narrow separations between the overlying aerosol and cloud tops.

*Specific comments: Page 2, Line 8-10: Potential edits: . . ..reduce cloud fraction (Hansen et al., 1997; Ackerman. . .) by reducing stability and relative humidity of the PBL, whereas . . . (Johnson et al., 2004; Sakaeda . . .)*

Thank you for the citation suggestions — they have been incorporated.

*Page 2, Line 22: add "for clouds with little precipitation" after cloud*

We have replaced "may suppress precipitation" with "may suppress drizzle" in line 22.

*Page 7, Line 5-12: Some of the conclusions seem to be drawn from Table 1. Add citation somewhere in this paragraph, or simply add the values of R2 to Figure 2.*

Conclusions drawn very specifically from Table 1 have been updated to refer readers to the precise rows being discussed in the following paragraph. The $R^2$ values for panels a)-e) in Figure 2 are provided in panel f), so the conclusions drawn in this paragraph are fully encapsulated within Figure 2's information. A specific reference to Figure 2f) for the $R^2$ values and Table 1 for the rest of the statistics has been added.

*Page 8, Line 1: add "(Fig. 2b)" after August*

Updated.

*Page 8, Line 11-12: I am not sure I understand why the sequence of flight maneuvers suggests direct instantaneous smoke cloud contact*

This would perhaps be better illustrated by the full time-height profile of the 4 September flight, but the RMP observations with no above cloud gap precede both CLD legs, although there's a brief cloud "dip" and some above cloud sampling before the straight-and-level cloud legs technically begin. We have clarified this in the text. The profiles that show potential gaps were further removed from the in-cloud legs (there were below cloud legs in between).

*Page 12, Line 18: Zhao et al., 2017 is not in the reference.*

"Zhao" is a typo — thank you for catching the error. Zhou et al. (2017) is cited on page 18, lines 26-28.

**RC2 (Reviewer 2):**

*Page 1, line 23: "Precipitation processes are also expected to obscure the relationship between above-cloud smoke and cloud properties in parts of the southeast Atlantic, although marine boundary layer carbon monoxide concentrations for two case study flights suggest that smoke entrainment history drove the observed differences in cloud properties for those days."*
*This sentence could be simplified for clarity; e.g.*
*"Precipitation processes can obscure the relationship between above-cloud smoke and cloud properties in parts of the southeast Atlantic, but marine boundary layer carbon monoxide concentrations for two case study flights suggest that smoke entrainment history drove the observed differences in cloud properties for those days."*

Thank you for the suggestion — it has been incorporated.

*Page 4, line 7: "... in situ data used is available at ..."*
*replace with*
*"... in situ data used are available at ..."*

Fixed.

*Page 5, line 15: "Only data from liquid clouds in the MBL (successful liquid cloud phase retrievals with effective temperatures below 280 K) ..."*
*Please specify what effective temperature is and adjust wording so that it is clear what "below" means (temperature or altitude).*

Clarified that the effective cloud top temperature must be warmer than 280 K and the cloud phase and effective cloud top temperature are retrieved from SEVIRI along with effective radius and cloud optical thickness.

*Page 7, line 18: "This indicates that variability in aerosol properties immediately above the MBL has little immediate impact on the microphysics of the clouds below."*
*This seems to follow from Table 1, Row 7, rather than from the context of this sentence.*

The whole paragraph has been updated to provide clearer references to Figure 2f and the appropriate rows of Table 1. The point of the line in question can be made equally well by comparing the coefficients of determination in rows 6 and 7 of Table 1 or by looking at the differences in the regression coefficients between the AC and BC values in rows 4 and 5.

*Page 8, line 26: "Observed CO (Fig. 6) is qualitatively consistent with the WRF-AAM output, with MBL average CO values on 31 August considerably above those from 4 September and among the highest seen during the deployment (all other flights shown in thin grey lines)."*
*It seems a bit too much to ask the reader to compare Fig. 5 b, c (WRF-AAM data) with Fig. 6 (ORACLES data) and to come to this conclusion. Please add, e.g., a time averaged vertical profiles of CO from Fig. 5 b, c to Fig. 6 to enable the comparison.*

The CO data from the model is only that specifically tagged to biomass burning emissions, versus the observations which are bulk measures of CO. Thus, the values are not directly comparable, which is why we specify the agreement is qualitative. However, it becomes quite clear from looking at the two figures that the day with higher model BB-tagged CO also has considerably higher observed CO.

*Page 12, line 7: "For instance, because the climatological MBL flow is southerly in the SEA, an instantaneous snapshot of smoke-cloud contact in the southern reaches of the domain may underestimate the microphysical effects by not accounting for their upstream manifestation. Similarly, apparently 'clean' cases in the northern part of the domain may have been polluted downstream, complicating efforts to compare 'mixed' and 'unmixed' statistics."*
*This sentence is a bit confusing. It seems to make more sense if "upstream" and "downstream" are switched.*

Thank you for this comment — the sentence has been significantly clarified by using geographic specifiers in lieu of upstream/downstream.

[revised manuscript text omitted]

---

## Author Response (AR2)

**Authors' Response to Co-editor**

Thank you for these clarifying suggestions. Specific changes and responses to the co-editor's comments are outlined point-by-point below, with comments in red italicized font and the authors' response in black.

In addition to the changes outlined below, we have performed a final check for typos, terminology, grammar, sentence structure, spelling, etc., and made minor adjustments as instructed in the notification email.

*1. Page 7, Lines 19-20. it is true that SO4 contributes to the CCN from both the natural marine and polluted sources. it would be good to add a brief discussion of the sources here. For example, marine source can be through the DMS oxidation. Then what is the polluted source of SO4 in the SE Atlantic? Are the smoke fires a dominating source? through fire emitted SO2 oxidation?*

Further explanation that the marine sulfate is from both sea spray and oxidation of DMS and the continental sulfate is a mix of (presumably some) industrial sources and (in large part) the biomass burning. A reference (Formenti et al. 2003) to SAFARI-2000 data showing that the oxidation of $SO_2$ from the burning is sufficient to account for "excess" sulfate over the southern African continent was added.

*2. Page 8, Line 15 "but do not account for..." this part of the sentence is hard to understand. what do you mean "high SO4 values do not account for.."?*

We have clarified that although the sulfate differences themselves account for some of the difference in $N_d$ between the two days, and in particular the incredibly high $N_d$ seen in some of the 31 August profiles, the rBC differences (and also the CO introduced later) are not explained by the sulfate story and thus something larger must be going on to explain the overall difference in MBL pollution.

*3. Page 11, Line 11, "has taken the place of CCNft from earlier". this part of the sentence is not complete. from earlier equation (6)?*

We have ended the previous equation with a period and started this qualifier as a new independent sentence for clarity.

*4. Page 12, Lines 9-10, "similar vertical distributions of BBA". It is only similar above clouds, not beneath clouds.*

Clarified.

*5. Page 12, Line 23, "-7-8 W m-2" should be "-7 to -8 W m-2".*

Fixed.

*6. Figure 4. in (a), I don't see the vertical marker position indicating the cloud top height. Also, better to note SEV inside (b) representing "SEVIRI"*

The caption has been updated for (a) to clarify that the horizontal position of the markers is $N_d$ and the vertical position is cloud top height and for (b) to clarify that "SEV" is representing SEVIRI.

*7. Figure 5. what is the dashed black line in (a)?*

The caption has been updated to clarify that the black dashed line in (a) is the routine flight track, included for reference.

[revised manuscript text omitted]